# Analysis of the Spatial and Temporal Evolution of China's Energy Carbon Emissions, Driving Mechanisms, and Decoupling Levels

Jingyi Ji [1][ID], Chao Li [1], Xinyi Ye [1], Yuelin Song [2] and Jiehua Lv [1],*

[1] College of Economics and Management, Northeast Forestry University, Harbin 150040, China; 13230506725@163.com (J.J.); 18972494724@163.com (C.L.); 2959271352@nefu.edu.cn (X.Y.)

[2] College of Forestry, Northeast Forestry University, Harbin 150040, China; 17726732009@163.com

* Correspondence: lvjiehua2004@126.com

**Abstract:** Excessive carbon emissions will cause the greenhouse effect and global warming, which is not conducive to environmental protection and sustainable development. In order to realize the goal of "carbon peak and carbon neutrality" as soon as possible, this paper utilizes the methodology provided by the IPCC to measure the carbon emissions and carbon intensity of China's energy consumption. The classification method of carbon emission and the kernel density function method are used to explore the spatial and temporal evolution of regional carbon emissions. Based on the Log Mean Divided Index (LMDI) method, the drivers of China's energy carbon emissions are measured. Based on the Tapio index function and the catch-up decoupling model, the decoupling status of Chinese provinces and the development gap with the benchmark provinces are examined. The results show that (1) China's total energy carbon emissions show a "rising-declining-rising" trend from 2005 to 2021, and reach the first peak in 2013, totaling 1,484,984.406 million metric tons. China's Hebei, Shanxi, and Shandong provinces have the highest energy carbon emissions. (2) China's energy carbon emissions are influenced by multiple factors, and the contribution of each factor to energy carbon emissions is in the following order: economic development effect > energy intensity effect > energy structure effect > population size effect. (3) China's catch-up provinces develop their economies at the expense of the environment and energy consumption.

**Keywords:** sustainable development; carbon peak and carbon neutrality; carbon emissions; Log Mean Divided Index (LMDI); catching-up decoupling; Tapio decoupling index



## 1. Introduction

With the rapid growth of the global population and the rapid increase in the consumption of natural resources, the risks and pressures on the global life system are increasing, which has led to a high incidence of ecological problems, including global warming due to the massive emission of greenhouse gases [1]. Although global carbon emissions have experienced a short-term decline due to the influence of the COVID-19 pandemic [2], the urgency and importance of addressing the challenge of climate change have not changed. According to the International Energy Agency (IEA), global energy-related carbon dioxide ($CO_2$) emissions will reach more than 36.88 billion metric tons in 2022, with China leading the world with 10.2 billion metric tons of $CO_2$ emissions [3]. China leads the world with 10.2 billion metric tons of carbon dioxide emissions. As a responsible power, China has made an absolute commitment at the UN General Assembly to "strive to reach peak carbon emissions by 2030 and achieve carbon neutrality by 2060" [4]. In its 14th Five-Year Plan, China has incorporated the goal of "dual carbon" into the overall layout of ecological civilization construction. Therefore, in recent years, China's carbon emission reduction efforts have been increasing, and the energy transition and energy revolution are being

promoted in multiple dimensions. However, due to the vastness of China, there are differences in energy and carbon emissions in different regions, which poses a challenge to the formulation and implementation of carbon emission reduction strategies. Therefore, it is of great significance to study China's energy carbon emissions at the provincial scale in order to realize energy saving and emission reduction and formulate scientific emission reduction policies in China.

In recent years, scholars have carried out various researches in the field of energy consumption and its carbon emissions. At present, the research of scholars at home and abroad on the issue of global carbon emissions mainly focuses on the measurement of carbon emissions, the decomposition of influencing factors, and the relationship between carbon emissions and economic growth.

For carbon emission measurement, most scholars use the baseline methodology provided in the 2006 IPCC Guidelines for Greenhouse Gas Emission Inventories published by the IPCC to estimate $CO_2$ emissions. For example, Roberta Quadrelli and Sierra Peterson (2007) [5] used the IPCC methodology to measure global carbon emissions and examined the drivers of carbon emissions. Pan et al. (2021) [6] measured the carbon emissions of 11 provinces and cities in eastern China, including Beijing, Tianjin, Hebei, and Jiangsu, based on the IPCC methodology, and analyzed the future trend of carbon emissions in the eastern region. Nonini, L. et al. (2022) [7] calculated carbon stocks in the Italian Central Alps case study area according to the 2006 IPCC guidelines. Chen et al. (2023) [8] used the IPCC carbon emission factor method to calculate land use carbon emissions and quantitatively analyze and assess the temporal characteristics of carbon emissions. These studies provide important references for further exploration of carbon emission control.

In the study of carbon emission-influencing factors, scholars have used Kaya's constant equation, the STIRPAT model, and the logarithmic mean Diels' index method (LMDI) index decomposition method. Japanese professor Yoichi Kaya (1989) [9] was the first to propose Kaya's constant equation, which revealed the effects of population, per capita GDP, energy intensity, etc, on carbon emissions. Bo Jiang (2020) [10] used the STIRPAT model to evaluate the degree of influence of major factors such as affluence, energy consumption intensity, and industrial structure on carbon emissions in the three northeastern provinces. JinHua Liu (2022) [11] based on the LIMID decomposition model to identity the influencing factors of carbon emissions are decomposed into economic level, population size, energy intensity, etc., and the potential and countermeasures for carbon emission reduction in China are explored on the basis of scenario analysis. Jiang, Q. et al. (2023) [12] utilized the energy and carbon emission data of the industrial sector in Fujian Province from 2005–2019 and applied the LMDI decomposition method to decompose the carbon emission drivers of each industry. Miskinis, V. (2023) [13] used the Log Mean Divided Index (LMDI) methodology to assess the impact of changes in the number of employees, labor productivity, energy intensity, RES deployment, and emission intensity on GHG emission reductions in Esmetric tonia, Latvia, and Lithuania, as well as in the EU-27.

In the study of the relationship between carbon emissions and economic growth, academics mainly use the decoupling model to study the relationship between the two [14]. Refining the decoupling index system and constructing a decoupling index system that contains eight cases such as strong decoupling and weak negative decoupling. Qi et al. (2015) [15] used the Tapio decoupling model to examine the relationship between economic growth and total carbon emissions, per capita carbon emissions, and carbon intensity in six central provinces. Zhao et al. (2022) [16] used the Tapio decoupling model to analyze that carbon emissions and economic growth in Northeast China are mainly weakly decoupled. Wang et al. (2023) [17] used the Tapio decoupling model to explore the decoupling relationship between China's overall economy and China's provinces' economic development and carbon emissions, respectively. Li, X.-Y. (2023) [18] explored the decoupled state of China's transportation industry from 2000 to 2020 with the Tapio model.

In summary, existing studies have produced rich results covering the analysis of factors influencing national, regional, and provincial carbon emissions and the examination

of overall decoupling effects. However, most of the literature on decoupling stays at the static scale of decoupling, and few articles have examined the dynamic process of carbon emission reduction catching up with decoupling. Compared with previous studies, the important contributions of this paper are reflected in the following: first, calculating the energy carbon emissions and carbon emissions intensity, and analyzing the spatial distribution and regional differences of the two at the same time. The Gaussian kernel density function is constructed so as to show the dynamic evolution law of energy carbon emission more intuitively. Second, the log-mean Diels method is used to decompose the driving factors of China's energy carbon emissions. This method has a better robustness test than other factor decomposition methods, eliminates computational residuals, and makes the decomposition results more accurate. Third, it analyzes the dynamic history of catching up and decoupling of China's energy carbon emission reduction and measures the gap in economic level and carbon emission level between catching up provinces and benchmark provinces, so as to be more targeted in proposing emission reduction policies. Fourth, we analyze the evolutionary characteristics and decoupling status of China's carbon emissions, aiming to provide quantitative support for China's carbon emission reduction policies and directional guidance for achieving the goals of carbon peaking and carbon neutrality.

## 2. Materials and Methods

### 2.1. Carbon Emission Accounting Methods

In order to more objectively measure the carbon emission level of Chinese provinces, this paper uses energy carbon emission and energy carbon emission intensity together to evaluate the dynamic changes and regional differences of China's energy carbon emission.

### 2.1.1. Accounting for Energy Carbon Emissions

For accounting for energy carbon emissions, different data types and accounting methods produce different results [19]. Based on the characteristics of China's energy consumption, this paper adopts the IPCC method to account for China's fossil energy consumption, including nine energy types: raw coal, coke, crude oil, gasoline, kerosene, diesel fuel, fuel oil, natural gas, and liquefied petroleum gas. In addition to this, this paper also accounts for China's electricity consumption in terms of carbon emissions. The calculation formula is as follows:

$$C = Q_f + Q_e \tag{1}$$

where $Q_f$ is the carbon emissions from the nine fossil energy sources, and $Q_e$ is the carbon dioxide emissions from electricity consumption [20].

According to the methodology provided by the IPCC for calculating carbon emissions from fossil energy sources, carbon emissions are calculated by multiplying energy consumption by a $CO_2$ emission factor with the following formula:

$$Q_f = \sum_{i=1}^{9} E_i C_i = \sum_{i=1}^{9} E_i (NCV_i \times EF_i \times COF_i \times \frac{44}{12}) \tag{2}$$

where $E_i$ is the consumption of the *i*th energy source, $C_i$ is the combined $CO_2$ emission factor, and $NCV_i$ (average low level heat generation), $EF_i$ (carbon content), $COF_i$ (carbon oxidation rate) and the molecular ratio of $CO_2$ to C 44/12 are multiplied to obtain $C_i$ [21].

The $CO_2$ emissions from electricity for each year need to be calculated by multiplying the electricity consumption by the carbon emission factor for electricity for that year, using the formula:

$$Q_e = W\lambda \tag{3}$$

where W is the electricity consumption, and $\lambda$ is the provincial electricity carbon emission factor for the year.

### 2.1.2. Energy Carbon Intensity Accounting

Energy carbon intensity refers to the energy carbon emissions consumed per unit of economic output, generally expressed as energy carbon emissions per unit of GDP, to reflect the efficiency of regional energy utilization. The formula for calculating energy carbon intensity is:

$$CI = \frac{C}{GDP} \tag{4}$$

where CI is the energy carbon intensity; GDP is the gross national product.

### 2.2. Kernel Density Estimation

In order to more intuitively reveal the dynamic evolution characteristics of energy carbon emissions in China, this paper applies non-parametric estimation of kernel density estimation to study the overall spatial differences and dynamic evolution trends of China's inter-provincial energy carbon emissions in the period of 2005–2021, and to measure the degree of agglomeration and dispersion through the height and width of the wave crests [22]. The calculation formula is as follows:

$$f(x) = \frac{1}{30} \sum_{i=1}^{30} K\left(\frac{X_i - \mu}{h}\right) \tag{5}$$

where $X_i$ is the energy carbon emissions of province i (i = 1, 2, ..., 30); h is the bandwidth; and K is the Gaussian kernel function.

### 2.3. Classification of Carbon Emissions

The average energy carbon emissions and the average energy carbon intensity of all provinces in China are used as the standard. Those higher than the average energy carbon emissions are considered high energy carbon emissions, while those lower than the average energy carbon emissions are considered low energy carbon emissions. Accordingly, China's inter-provincial energy carbon emissions are categorized into four groups [23], namely H-H (high energy carbon emissions—high energy carbon emissions intensity), L-H (low energy carbon emissions—high energy carbon emissions intensity), H-L (high energy carbon emissions—low energy carbon emissions intensity), and L-L (low energy carbon emissions—low energy carbon emissions intensity).

### 2.4. Log Mean Divided Index (LMDI) Exponential Decomposition

Energy, economy, population, etc, are usually recognized as important factors affecting carbon emissions. Quantifying the specific degree of influence and contribution of different factors on energy carbon emissions is crucial to the scientific formulation of emission reduction policies. Drawing on the methodology of Liu's et al. [24] method, we establish a log-mean Diels' index equation to analyze the influence of energy structure, energy intensity, economic development, and population size on China's energy carbon emissions. The specific equations are as follows:

$$C = \sum_i \sum_j C_{ij} = \sum_i \sum_j \frac{C_{ij}}{E_{ij}} \times \frac{E_{ij}}{E_i} \times \frac{E_i}{G_i} \times \frac{G}{P} \times P = \sum_i \sum_j CI \times ES \times EI \times GP \times P \tag{6}$$

$$\Delta C = C_t - C_0 = \Delta CCI + \Delta CES + \Delta CEI + \Delta CGP + \Delta CP \tag{7}$$

where $G_{ij}$ is the carbon emissions from fossil fuels in industry i,j; $E_{ij}$ is the energy consumption of fossil fuels in industry j in industry i; $E_i$ is the total energy consumption of industry i; $G_i$ is the GDP of industry i, G is the total GDP; P is the population at the end of the year; CI, ES, GP, and CP are the carbon emission factor, energy structure, energy intensity, economic growth, and population size, respectively; $\Delta C$ is the total effect; $C_t$ and $C_0$ are the target year and base year carbon emissions, $\Delta CCI$, $\Delta CES$, $\Delta CEI$, $\Delta CGP$, and $\Delta CP$ are the effects of

each influencing factor on carbon emissions, respectively. The formula for calculating the effect of each influence factor is as follows:

$$\Delta C_x = \sum_i \sum_j L\left(C_{ij}^t, C_{ij}^0\right) \ln\left(\frac{x^t}{x^0}\right) \tag{8}$$

where: x is each of the above influencing factors; $\Delta C_x$ is the carbon emission effect of influence factor x; $L\left(C_{ij}^t, C_{ij}^0\right)$ is the weight; $C_{ij}^t$ and $C_{ij}^0$ are the carbon emissions from fossil fuels in industry j in the target year and base year, respectively; $x^t$ and $x^0$ are the values of the influencing factors in the target year and base year, respectively. Among them, the weighting formula is as follows:

$$L\left(C_{ij}^t, C_{ij}^0\right) = \left\{ \begin{array}{ll} \frac{C_{ij}^t - C_{ij}^0}{\ln C_{ij}^t - \ln C_{ij}^0}, & C_{ij}^t \neq C_{ij}^0 \\ C_{ij}^t \text{ or } C_{ij}^0, & C_{ij}^t = C_{ij}^0 \\ 0, & C_{ij}^t = C_{ij}^0 = 0 \end{array} \right\} \tag{9}$$

In order to facilitate the comparison, the relative contribution degree is used to describe the degree of influence of each effect on carbon emissions, based on the research method of Wang's et al. [25] research method, the relative contribution degree is used to describe the degree of influence of each effect on carbon emissions, with the following formula:

$$\theta = \frac{\Delta C_x}{\sum_x |\Delta C_x| \times 100\%} \tag{10}$$

$\theta$ is the relative contribution degree. $\theta > 0$, the influence factor has a promoting effect on carbon emissions, and the larger the value, the stronger the promoting effect; $\theta < 0$, the influencing factor has an inhibitory effect on carbon emissions, and the larger the absolute value, the stronger the inhibitory effect.

### 2.5. Tapio's Decoupling Index Model

The Tapio decoupling index model is a method of elasticity coefficient analysis constructed by Tapio in his study of the relationship between economic development in Europe, transportation capacity, and the $CO_2$. Different decoupling coefficients characterize different states of development, and the traditional decoupling index model is divided into eight categories. The Tapio decoupling index is used to dynamically observe the decoupling characteristics of variables, and more clearly reflects the relationship between each factor in terms of economic development and environmental stress [26]. Therefore, this paper chooses the Tapio model to construct the decoupling relationship between carbon emissions and economic growth in China. The decoupling relationship is modeled as:

$$DI = \frac{\%\Delta TC}{\%\Delta GDP} = \frac{\Delta TC/TC}{\Delta GDP/GDP} = \frac{(TC_{t+1} - TC_t)/TC_t}{(GDP_{t+1} - GDP_t)/GDP_t} \tag{11}$$

DI is the decoupling elasticity index; $\%\Delta GDP$ is the rate of change of gross regional product; $\%\Delta TC$ is the rate of change of carbon emissions. $TC_t$ and $TC_{t+1}$ are the energy carbon emissions in period t and period t + 1, respectively. $GDP_{t+1}$ and $GDP_t$ are the GDP in period t and t + 1. The classification of the results of the decoupling index is shown in Figure 1.

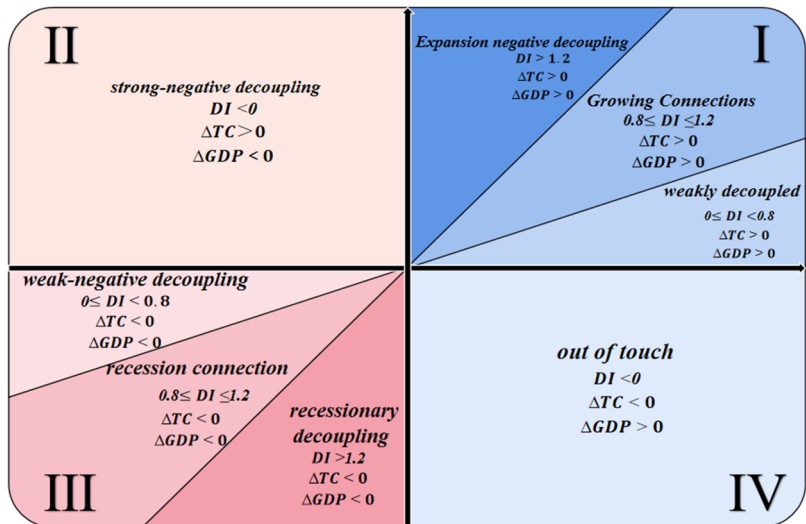

**Figure 1.** Division of decoupling states. I represents the division of weakly decoupling, growing connections, and expansion negative decoupling. II represents strong negative decoupling. III represents weak-decoupling, recession connection, recessionary decoupling. IV represents out of touch.

*2.6. Catch-Up Decoupling Model*

The Tapio decoupling index model portrays a comparison between the speed of economic development and the speed of energy and carbon emissions, which is a comparison of its own speed and does not reflect the gap between economic development and carbon emissions and other regions. Referring to Zhang et al. (2013) [27], the provinces with "good economic development and low carbon intensity" are defined as benchmark provinces. In order to describe the dynamic decoupling process of Chinese provinces catching up with the benchmark provinces, this paper constructs the following catching-up decoupling model based on the theoretical foundation of the Tapio decoupling coefficient model:

$$\mathrm{T}^Z_{it} = \frac{-\left[\left(CE^n_t - CE_{it}\right) - \left(CE^n_{t-1} - CE_{i,t-1}\right)\right] / \left(CE^n_{t-1} - CE_{i,t-1}\right)}{-\left[\left(PG^n_t - PG_{it}\right) - \left(PG^n_{t-1} - PG_{i,t-1}\right)\right] / \left(PG^n_{t-1} - PG_{i,t-1}\right)} = \frac{\Delta CE}{\Delta PG} \tag{12}$$

$\mathrm{T}^Z_{it}$ represents the catch-up decoupling elasticity index for province i in year $CE_{it}$ and $PG_{it}$ represent the carbon intensity and per capita GDP level of province i in year t, respectively; $CE^n$ and $PG^n$ represent the carbon intensity and per capita GDP level of the benchmark province, respectively. The classification criteria for catching up and decoupling are shown in Table 1.

**Table 1.** Criteria for categorizing catch-up decoupling.

| Catch-Up Type | Economic Growth Gap | Carbon Intensity Gap | Catch-Up Decoupling Elasticity Factor | Catch-Up Effect |
|---|---|---|---|---|
| Absolute catch-up decoupling (A) | $\Delta PG < 0$ | $\Delta CE > 0$ | $\left|\mathrm{T}^Z_{it}\right| > 1$ | Emission reduction catch-up is better than economic catch-up |
| Absolute catch-up decoupling (B) | | | $0 < \left|\mathrm{T}^Z_{it}\right| \le 1$ | Economic catch-up is better than emission reduction catch-up |
| Relative catch-up decoupling (A) | $\Delta PG < 0$ | $\Delta CE < 0$ | $0 < \left|\mathrm{T}^Z_{it}\right| \le 1$ $\left|\mathrm{T}^Z_{it}\right| > 1$ | Economic catch-up is better than emission reduction catch-up |
| Relative catch-up decoupling (B) | $\Delta PG > 0$ | $\Delta CE > 0$ | $0 < \left|\mathrm{T}^Z_{it}\right| \le 1$ $\left|\mathrm{T}^Z_{it}\right| > 1$ | Emission reduction catch-up is better than economic catch-up |

**Table 1.** *Cont.*

| Catch-Up Type | Economic Growth Gap | Carbon Intensity Gap | Catch-Up Decoupling Elasticity Factor | Catch-Up Effect |
|---|---|---|---|---|
| Failure to catch up with decoupled (A) | $\Delta PG > 0$ | $\Delta CE < 0$ | $\left|T_{it}^Z\right| > 1$ | Emission reduction lags behind economic growth |
| Failure to catch up with decoupled (B) | | | $0 < \left|T_{it}^Z\right| \leq 1$ | Economic growth lags behind emission reductions |

*2.7. Explanatory Variables Selection and Description*

The energy consumption, population, and economic data for China and each province were obtained from the China Statistical Yearbook 2006–2022, the China Energy Statistical Yearbook, as well as statistical yearbooks and official websites of statistical bureaus of each province (Hong Kong, Macao, Taiwan, and Tibet were not included due to missing data). The population data is based on the resident population, and the GDP data is converted to 2005 constant prices to exclude the effect of inflation. The discounted standard coal coefficient adopts the value provided in the General Rules for Calculating Comprehensive Energy Consumption. The data on average low-level heat generation, carbon content per unit calorific value (default value) and carbon oxidation rate are from the Guidelines for the Preparation of Provincial Greenhouse Gas Inventories (2011 Trial Version), and the average carbon dioxide emission factors for provincial power grids are from the Average Carbon Dioxide Emission Factors for China's Regional Power Grids in 2011 and 2012. The interpretation and sources of the indicators are shown in Table 2.

**Table 2.** Description of data sources for indicators.

| Norm | Yearbook Data Involved |
|---|---|
| energy structure | Energy consumption |
| energy intensity | Energy consumption, GDP |
| economic development | GDP |
| population size | population |
| NCV | Average low level heat generation |
| COF | carbon oxidation rate |
| $\lambda$ | Provincial grid average $CO_2$ emission factor |

**3. Results**

*3.1. Characteristics of Spatial and Temporal Changes in Energy Carbon Emissions*

Based on Equations (1)–(3), this paper accounts for the carbon emissions generated from the consumption of nine types of energy and electricity, namely, raw coal, coke, crude oil, gasoline, kerosene, diesel oil, fuel oil, natural gas, and liquefied petroleum gas, from 2005 to 2021. Due to the space limitation of the article, the specific accounting results of energy carbon emissions are put in Appendix A.

3.1.1. Time Evolution Characteristics

From 2005 to 2021, China's total energy carbon emissions showed a "rising-declining-rising" trend, reaching its first peak in 2013, with a total of 1,484,984,406 metric tons, and showing a decreasing trend from 2013 to 2015. As shown in Figure 2. This is related to China's implementation of the National 12th Five-Year Plan for Ecological Protection in 2013. Throughout 2015–2021, China's energy carbon emissions show a fluctuating and increasing trend, with a total of 1394,239,874 metric tons in 2015, growing at an average annual rate of 3.51%, reaching a maximum value during the study period in 2020, and reaching a maximum value in 2020, with an average annual growth rate of 3.51%. In 2020, it reached the maximum value of 1,605,656.017 million metric tons in the study period. Subsequently, the energy carbon emissions show a slow decreasing trend, and the level

of energy carbon emissions in 2021 is almost the same as that in 2019. This is because, in early 2020, China took stringent precautions to prevent the spread of a new crown outbreak. The short-term social shutdown had a huge impact on the domestic economy, while also curbing the growth of carbon emissions.

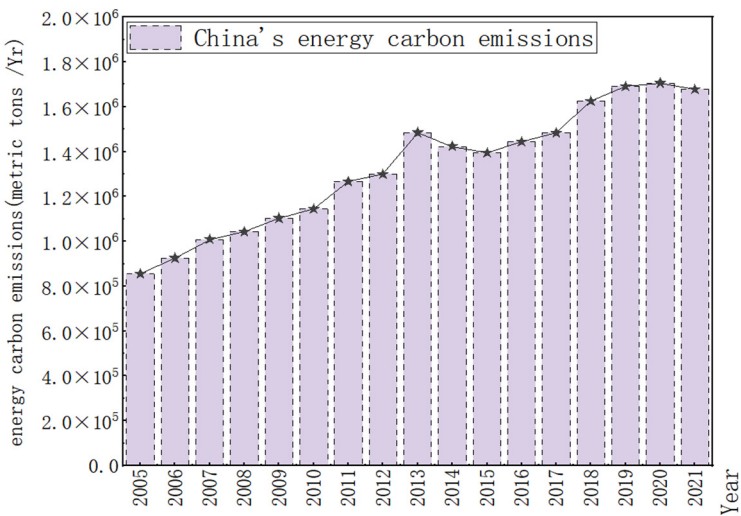

**Figure 2.** Evolutionary characteristics of China's energy carbon emissions, 2005–2021.

3.1.2. Characteristics of Spatial Evolution

Based on the energy carbon emission data of Chinese provinces in 2005, 2009, 2013, 2017, and 2021, with the help of ArcGIS 10.8, and combined with the natural breakpoint grading method, the energy carbon emissions of Chinese provinces were categorized into 5 levels (in metric tons): level 1 [0, 10,000], level 2 [10,000, 50,000], level 3 [50,000, 100,000], level 4 [100,000, 150,000], level 5 [150,000, $+\infty$].

As shown in Figure 3, the spatial pattern of energy carbon emissions in China's provinces has changed considerably from 2005 to 2021, with a decrease in low-value regions and an increase in high-value regions. Specifically, a total of three provinces were in the level 3 carbon emission range in 2005, namely Hebei Province, Shanxi Province, and Shandong Province, with high energy carbon emissions. The reason for this is that Shanxi Province is China's energy base, particularly rich in coal resources, and its economic growth relies mainly on the large consumption of fossil energy, so energy carbon emissions are relatively high. Hebei Province and Shandong Province are important heavy industry bases in China, and the development of high heavy industry requires the consumption of large amounts of energy, which leads to energy carbon emissions at the forefront of the country. Qinghai Province, Ningxia Hui Autonomous Region, and Chongqing Municipality are in Level 1, with the lowest energy carbon emissions. This is mainly because they are in the western region of China, where the level of economic development is low and the level of energy consumption is relatively low. The rest of the provinces are in level 2 with [10,000, 50,000] million metric tons. In 2009, compared with 2005, five provinces, Inner Mongolia, Liaoning, Henan, Jiangsu, and Guangdong, evolved from carbon emission level 2 to level 3, Ningxia and Chongqing evolved from level 1 to level 2, and the energy carbon emission level of the rest of the provinces remained unchanged. This suggests that China's energy carbon emissions is on an upward trend, with most provinces developing their economies at the expense of fossil energy consumption. The spatial pattern of China's energy carbon emissions in 2013 is roughly the same as that of 2009, with the exception of Shandong province, where the energy mix has evolved from Tier 3 to Tier 4. Shandong province's energy structure is dominated by high-carbon fossil energy with 88% of fossil energy, the highest in China, which has led to Shandong province becoming the top province in China in terms of energy carbon emissions. In 2017, energy carbon emissions in Xinjiang and Shaanxi increased more significantly, shifting from Tier 2 to Tier 3. This is because after the

promulgation of the 13th Five-Year Plan for the Development of the Western Region, the economies of western provinces such as Xinjiang and Shaanxi have developed significantly, but technological advances have not been able to keep up with economic development, so their economic development has been achieved by consuming large amounts of fossil energy. In 2021, the evolution of carbon emissions in China's northern provinces is more dramatic, with Inner Mongolia, Hebei, and Shanxi entering tier 4, Hebei and Shanxi enter level 4, and Shandong reaches level 5, the highest level within the study, i.e., annual energy carbon emissions greater than 150,000,000 metric tons. The rest of the provinces have no significant change in their energy carbon emission levels compared to 2017.

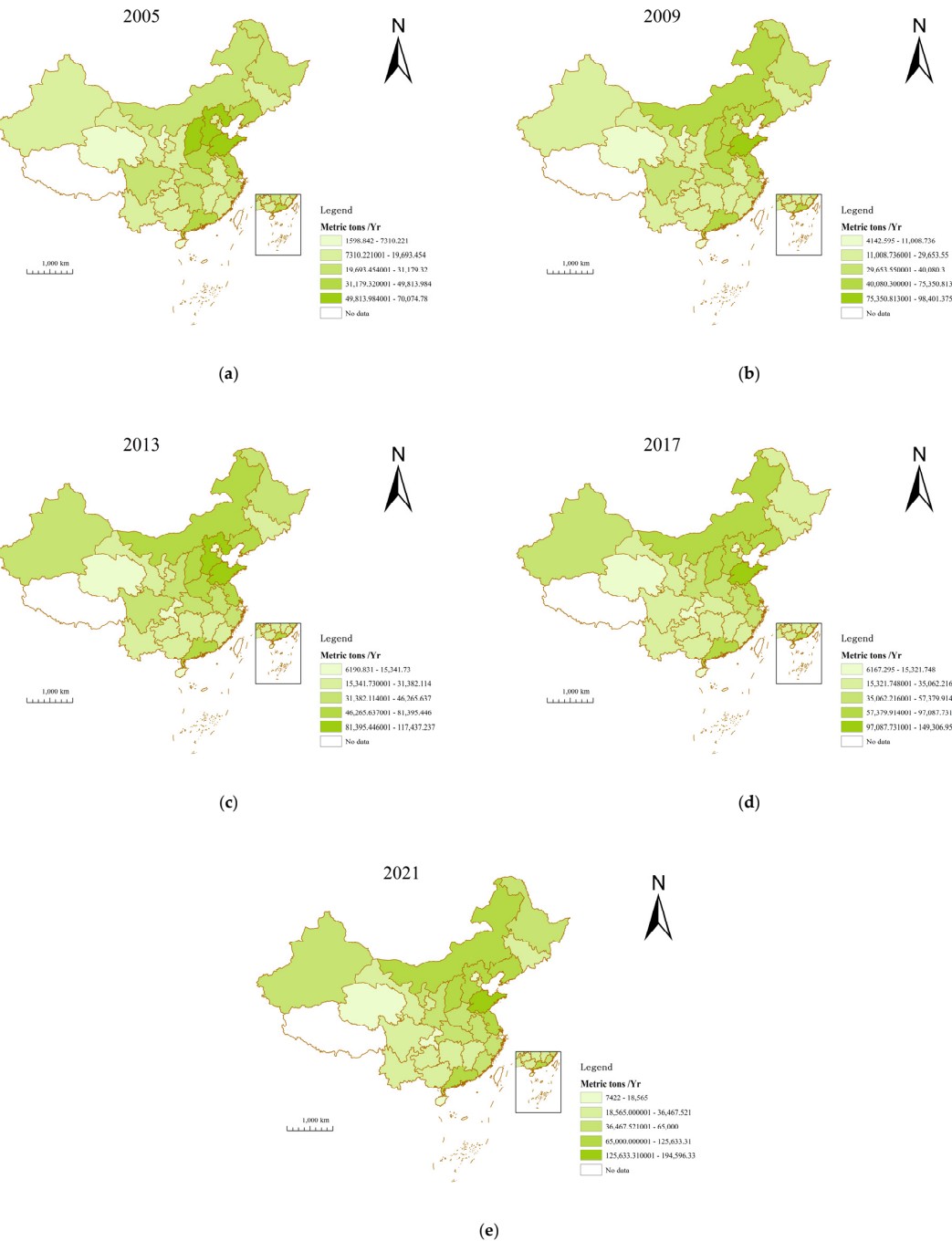

**Figure 3.** Spatial evolution of energy carbon emissions in China, (**a**) Energy Carbon Emissions by Province in China, 2005; (**b**) Energy Carbon Emissions by Province in China, 2009; (**c**) Energy Carbon Emissions by Province in China, 2013; (**d**) Energy Carbon Emissions by Province in China, 2017; (**e**) Energy Carbon Emissions by Province in China, 2021.

### 3.1.3. Analysis of the Dynamic Evolution of Disequilibrium

In order to further study the differences and dynamic evolution trends of energy carbon emissions in Chinese provinces, based on the previous analysis of the temporal and spatial characteristics of energy carbon emissions, using Equation (5), a three-dimensional Gaussian kernel density curve is constructed, as shown in Figure 4. With regard to the position of the kernel density function, from 2005 to 2013, the density distribution interval shows an overall rightward flat trend, indicating that China's energy carbon emissions were continuously rising from 2005 to 2013, which is consistent with the evolutionary law of China's energy carbon emissions in the time dimension; from 2013 to 2019, the interval shifts to the left, indicating that the intensity of carbon emissions has declined, and from 2019 to 2021, the interval has a tendency to move to the right. In terms of the kernel function's kurtosis, a double-peak pattern is shown in 2010, with China's inter-provincial energy carbon emissions concentrating in the range of $5 \times 10^4$ million metric tons as well as $10 \times 10^4$ million metric tons, with more obvious polarization. In terms of the distribution pattern, the right trailing tail shows an extension trend, indicating that the differences in China's inter-provincial energy carbon emissions are gradually expanding.

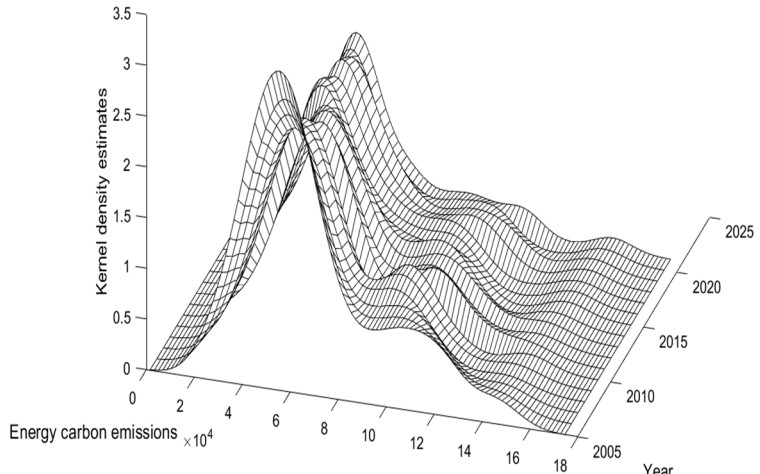

**Figure 4.** Kernel density function for the dynamic evolution of energy carbon emissions.

### 3.2. *Classification of Carbon Emissions*

According to Equation (4), China's energy carbon emission intensity was calculated. Linking with the previous energy carbon emissions, the carbon emissions were divided into four categories, which were H-H (high energy carbon emissions—high energy carbon intensity), L-H (low energy carbon emissions—high energy carbon intensity), H-L (high energy carbon emissions—low energy carbon intensity), and L-L (low energy carbon emissions—low energy carbon intensity).

The classification of China's inter-provincial energy carbon emissions from 2005 to 2021 is shown in Table 3. During the study period, Hebei, Shanxi, Inner Mongolia, and Liaoning are all of type H-H, indicating that these provinces need to take into account the aspects of energy transition and technological innovation to accelerate energy transition and innovate their economic development methods. The carbon emission types of Jiangsu, Zhejiang, and Guangdong are relatively stable and have been of type H-L. The economic development of these provinces is at a leading level in China, and they are strong economic provinces, but the efficiency of energy use is not high, so these provinces should focus on developing and utilizing cleaner energy sources in the future, and reduce their dependence on fossil energy sources. Provinces in the L-H type, such as Jilin, Guizhou, Ningxia, Qinghai, etc., should improve the level of technological innovation, develop their economies, and increase the efficiency of resource utilization.

**Table 3.** Classification of energy carbon emissions.

| Particular Year | H-H | H-L | L-H | L-L |
|---|---|---|---|---|
| 2005 | Hebei, Shanxi, Inner Mongolia, Liaoning, Heilongjiang, Shandong, Henan | Jiangsu, Zhejiang, Hubei, Guangdong | Jilin, Guizhou, Yunnan, Shaanxi, Gansu, Qinghai, Ningxia, Xinjiang | Beijing, Tianjin, Shanghai, Anhui, Fujian, Jiangxi, Hunan, Guangxi, Hainan, Chongqing, Sichuan |
| 2013 | Hebei, Shanxi, Inner Mongolia, Liaoning, Shaanxi, Xinjiang | Jiangsu, Zhejiang, Shandong, Henan, Guangdong | Jilin, Heilongjiang, Guizhou, Yunnan, Gansu, Qinghai, Ningxia | Beijing, Tianjin, Shanghai, Anhui, Fujian, Jiangxi, Hubei, Hunan, Guangxi, Hainan, Chongqing, Sichuan |
| 2021 | Hebei, Liaoning, Shanxi, Shaanxi, Xinjiang, Inner Mongolia, Shandong | Jiangsu, Zhejiang, Guangdong, Henan | Jilin, Heilongjiang, Guizhou, Hainan, Ningxia, Gansu, Tianjin, Qinghai | Beijing, Anhui, Hunan, Chongqing, Fujian, Hubei, Sichuan, Yunnan, Shanghai, Jiangxi, Guangxi |

### 3.3. Analysis of the Driving Mechanism of China's Energy Carbon Emissions

With the help of existing research results [28,29] and China's actual situation, this paper studies the driving mechanism of energy carbon emission from four dimensions: economic development, energy intensity, energy structure, and population size. Using Equations (6)–(9), the driving mechanism of energy carbon emission in China is decomposed. The cumulative contribution values of economic development effect and population size effect to China's energy carbon emissions are 1,409,807.68 million metric tons and 978,539.90 million metric tons, respectively, and both of them are positive factors promoting the increase of energy carbon emissions, and the economic development effect contributes more values. The cumulative contribution values of energy structure and energy intensity to China's energy carbon emissions are −151,967.14 million metric tons and −704,101.50 million metric tons, respectively, which are negative factors inhibiting energy carbon emissions. Based on Equation (10), the contribution of each driver to China's energy carbon emissions is calculated and plotted as a heat map based on the contribution of the four drivers, as shown in Figure 5.

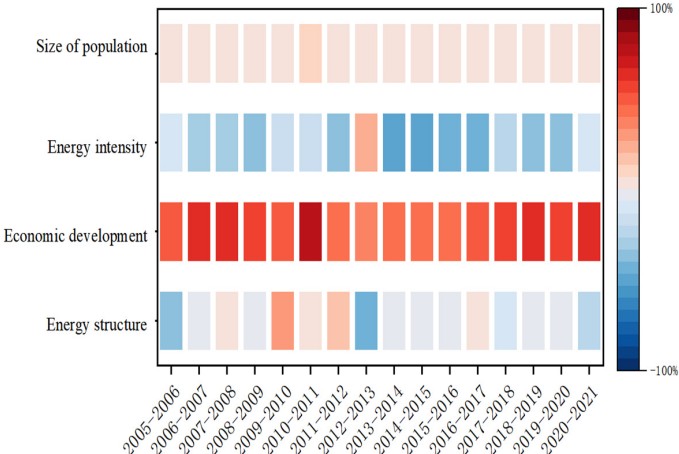

**Figure 5.** Contribution of each driver to China's energy carbon emissions.

From the results of Figure 4, the contribution rate of each factor to energy carbon emission is ranked as follows: economic development effect > energy intensity effect > energy structure effect > population size effect, and the cumulative contribution rates are 863.92%, −446.43%, −81.96%, and 58.23%, respectively.

Firstly, the economic development effect is the most important driver of China's rising energy and carbon emissions.

From 2005 to 2021, the overall contribution of economic development effect to China's energy carbon emissions was positive, and it was the most important driver of China's energy carbon emissions, and the contribution of economic growth effect was 74.19% in 2011, reaching the maximum value. During the "12th Five-Year Plan" period (2011–2015), the driving role of economic growth factors gradually weakened, but it is still the most important factor contributing to the growth of carbon emissions. This is related to China's economic development, after the release of the "12th Five-Year Plan", China's economy has changed from high-speed development to high-quality development [30], and energy carbon emissions have slowed down as a result.

Secondly, population size effect is one of the factors contributing to energy and carbon emissions.

The contribution rate of population scale effect is relatively stable during the study period, remaining at 1–6%. Due to the small change in the natural population growth rate and the popularization of the concept of environmental protection, it is less likely that the effect of population size on carbon emissions will increase in the future. This is consistent with the findings of Wang Quiet et al. (2023) [17].

Thirdly, energy intensity presents an inhibitory effect on carbon emissions.

During the study period, except for 2013, energy intensity presents an inhibitory effect on carbon emissions. The inhibitory effect of energy intensity on carbon emissions has gradually weakened in the past three years, indicating that China still needs to improve energy efficiency and realize green and low-carbon development of the economy.

Fourthly, Energy structure has a small impact on carbon emissions.

The contribution of energy structure to carbon emissions is both positive and negative, but overall, the factor is negatively inhibiting the growth of carbon emissions. However, from the point of view of the contribution rate, the factor's role in controlling carbon emissions is relatively weak.

### 3.4. Decoupling of Carbon Emissions from Economic Development at the Inter-Provincial Level in China

3.4.1. Overall Decoupling of Carbon Emissions from Economic Development at the Inter-Provincial Level

Based on Equation (11), the decoupling index was calculated for each province in China from 2005 to 2021. According to the characteristics of the decoupling relationship, the period 2005–2021 is divided into five time periods for its stage analysis, and the specific results are shown in Table 4. In the time period of 2005–2008, there are 26 provinces with weak decoupling between energy carbon emissions and economic development in China, namely Beijing, Tianjin, Hebei, etc; Hainan Province shows negative decoupling with expansion, and Chongqing and Qinghai are connected with growth. During 2009–2012, the number of provinces with weak decoupling is reduced to 22, and the number of provinces with growth is 4, namely Inner Mongolia Aumetric tonomous Region, Shanxi, Hainan, Jiangsu, and 3 provinces in Guangxi, Ningxia, and Xinjiang show negative decoupling with expansion. Hainan, and Jiangsu, and three provinces showing expansion negative decoupling in Guangxi, Ningxia, and Xinjiang. In 2013–2016, the number of provinces showing decoupling gradually increased, with 12 provinces showing strong decoupling and 16 provinces showing weak decoupling. In this time domain, no provinces showed expansion-negative decoupling. In 2017–2019, China's interprovincial mainly showed weak decoupling, strong decoupling, expansion-negative decoupling, and growth-connecting 4 bells decoupling. Compared with 2017–2019, the type of decoupling becomes single in 2020–2021, presenting only 2 patterns of weak decoupling and growth connection. The

above provinces showing decoupling between energy carbon emissions and economic development development can be categorized into 2 types. The first category is economically developed with a relatively low share of value-added of the energy industry, such as Beijing and Tianjin. The second category is economically underdeveloped provinces such as Henan and Sichuan, which have achieved the dual purpose of economic growth and environmental protection by following a new industrialization path. Provinces showing growth connection, such as Shandong, Hunan, and Guizhou, remain in the crude economic development mode, resulting in synchronized growth of energy and carbon emissions and the economy. Provinces such as Ningxia and Xinjiang show negative decoupling of expansion, which is attributed to the fact that in recent years, provinces in western China have vigorously developed coal and other energy-intensive industries in order to develop their economies at the expense of the environment, which has led to a faster rate of energy and carbon emissions than the rate of economic growth.

**Table 4.** Decoupling of energy carbon emissions from economic development at the inter-provincial level in China, 2005–2021.

|  | 2005–2008 | 2009–2012 | 2013–2016 | 2017–2019 | 2020–2021 |
|---|---|---|---|---|---|
| Beijing | weakly decoupled | out of touch | out of touch | weakly decoupled | weakly decoupled |
| Tianjin | weakly decoupled | weakly decoupled | out of touch | weakly decoupled | weakly decoupled |
| Hebei | weakly decoupled | weakly decoupled | weakly decoupled | Expansion negative decoupling | weakly decoupled |
| Shanxi | weakly decoupled | weakly decoupled | weakly decoupled | Expansion negative decoupling | weakly decoupled |
| Inner Mongolia | weakly decoupled | Growing Connections | weakly decoupled | Expansion negative decoupling | weakly decoupled |
| Liaoning | weakly decoupled | weakly decoupled | out of touch | Expansion negative decoupling | weakly decoupled |
| Jilin | weakly decoupled | weakly decoupled | out of touch | weakly decoupled | weakly decoupled |
| Heilongjiang | weakly decoupled | weakly decoupled | out of touch | Growing Connections | weakly decoupled |
| Shanghai | weakly decoupled | weakly decoupled | weakly decoupled | weakly decoupled | weakly decoupled |
| Jiangsu | weakly decoupled | Growing Connections | weakly decoupled | out of touch | weakly decoupled |
| Zhejiang | weakly decoupled | weakly decoupled | weakly decoupled | weakly decoupled | weakly decoupled |
| Anhui | weakly decoupled | weakly decoupled | weakly decoupled | weakly decoupled | weakly decoupled |
| Fujian | weakly decoupled | weakly decoupled | out of touch | Expansion negative decoupling | Growing Connections |
| Jiangxi | weakly decoupled | weakly decoupled | weakly decoupled | weakly decoupled | weakly decoupled |
| Shandong | weakly decoupled | weakly decoupled | Growing Connections | out of touch | Growing Connections |
| Henan | weakly decoupled | weakly decoupled | out of touch | out of touch | weakly decoupled |
| Hubei | weakly decoupled | weakly decoupled | out of touch | Expansion negative decoupling | weakly decoupled |
| Hunan | weakly decoupled | weakly decoupled | weakly decoupled | weakly decoupled | Growing Connections |
| Guangdong | weakly decoupled | weakly decoupled | weakly decoupled | Growing Connections | weakly decoupled |
| Guangxi | weakly decoupled | Expansion negative decoupling | out of touch | Growing Connections | weakly decoupled |
| Hainan | Expansion negative decoupling | Growing Connections | weakly decoupled | Expansion negative decoupling | weakly decoupled |
| Chongqing | Growing Connections | weakly decoupled | out of touch | weakly decoupled | Growing Connections |
| Sichuan | weakly decoupled | weakly decoupled | out of touch | weakly decoupled | weakly decoupled |

**Table 4.** *Cont.*

|  | 2005–2008 | 2009–2012 | 2013–2016 | 2017–2019 | 2020–2021 |
|---|---|---|---|---|---|
| Guizhou | out of touch | weakly decoupled | weakly decoupled | out of touch | Growing Connections |
| Yunnan | weakly decoupled | weakly decoupled | out of touch | Expansion negative decoupling | Growing Connections |
| Shaanxi | weakly decoupled | Growing Connections | weakly decoupled | Growing Connections | weakly decoupled |
| Gansu | weakly decoupled | weakly decoupled | out of touch | Growing Connections | weakly decoupled |
| Qinghai | Growing Connections | weakly decoupled | weakly decoupled | out of touch | Growing Connections |
| Ningxia | weakly decoupled | Expansion negative decoupling | weakly decoupled | Expansion negative decoupling | weakly decoupled |
| Xinjiang | weakly decoupled | Expansion negative decoupling | weakly decoupled | Expansion negative decoupling | weakly decoupled |

### 3.4.2. Analysis of Decoupling Transfers

The transfer of energy carbon emissions and economic development from 2005 to 2021 is shown in Figure 6. In terms of overall decoupling types, there are 4, 3, 3, 4, and 2 decoupling types in the five time periods, respectively, presenting a trend of decreasing decoupling richness; in terms of inter-provincial decoupling transfers, most of the provinces' decoupling status transfers are dominated by strong and weak decoupling transfers to each other, which suggests that most of China's provinces present a relatively stable decoupling trend between energy and carbon emissions and economic development. Hainan, Chongqing, and Guizhou show the same trend of decoupling transfer, and the three provinces are richer in the type of decoupling. The two provinces of Ningxia and Xinjiang show exactly the same change trend, which is the change pattern of "weak decoupling-expansion-negative decoupling-weak decoupling-expansion-negative decoupling-weak decoupling". This is due to the fact that Xinjiang and Ningxia are both located in Northwest China, with little difference in the level of development of economy, population, resources, and environment. Hua Ruixiang et al.(2023) [31] studied the decoupling effect of interprovincial carbon emissions and economic growth in China from 2015 to 2020, and found that although the annual decoupling status of each province shows a fluctuating trend of change, the overall view is dominated by the mutual transformation of strong decoupling and weak decoupling.

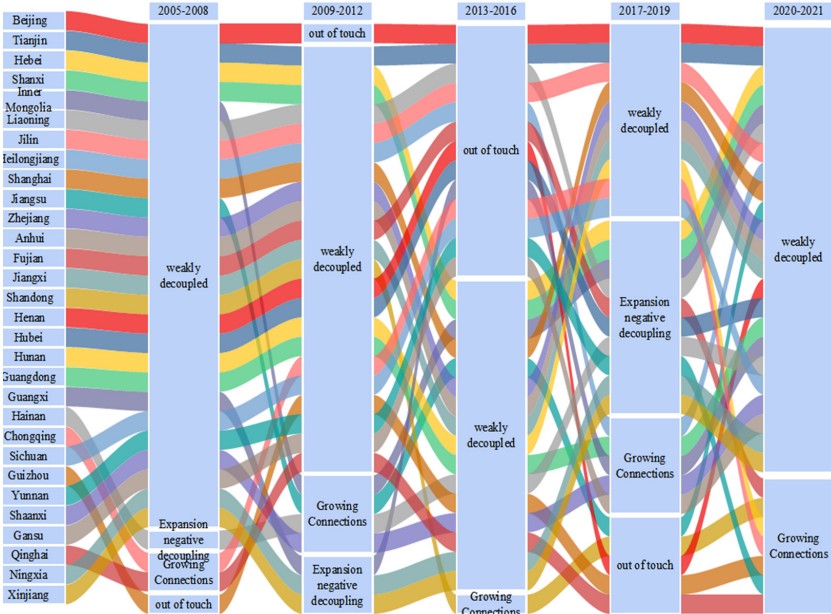

**Figure 6.** Shift in decoupling of energy carbon emissions from economic development, 2005–2021.

### 3.4.3. Catch-Up Decoupling Analysis

In order to conduct an in-depth study on the decoupling characteristics of China's energy and carbon emissions, a catching-up decoupling index model is constructed to describe the dynamic process of catching up from each province to the benchmark provinces. Referring to Zhang et al.'s (2013) [27] approach, five provinces with leading carbon intensity and per capita GDP indicators are selected. Through calculation and comparison, it is found that the top 5 are Beijing, Shanghai, Guangdong, Zhejiang, and Jiangsu. By averaging the values of these nine provinces year by year, a model province with excellent economic performance and carbon emissions is constructed. Using Equation (12), the results of the catch-up decoupling index calculations are categorized for each province. Figure 7 shows the process of catching up and decoupling provinces from the model province from 2005 to 2021.

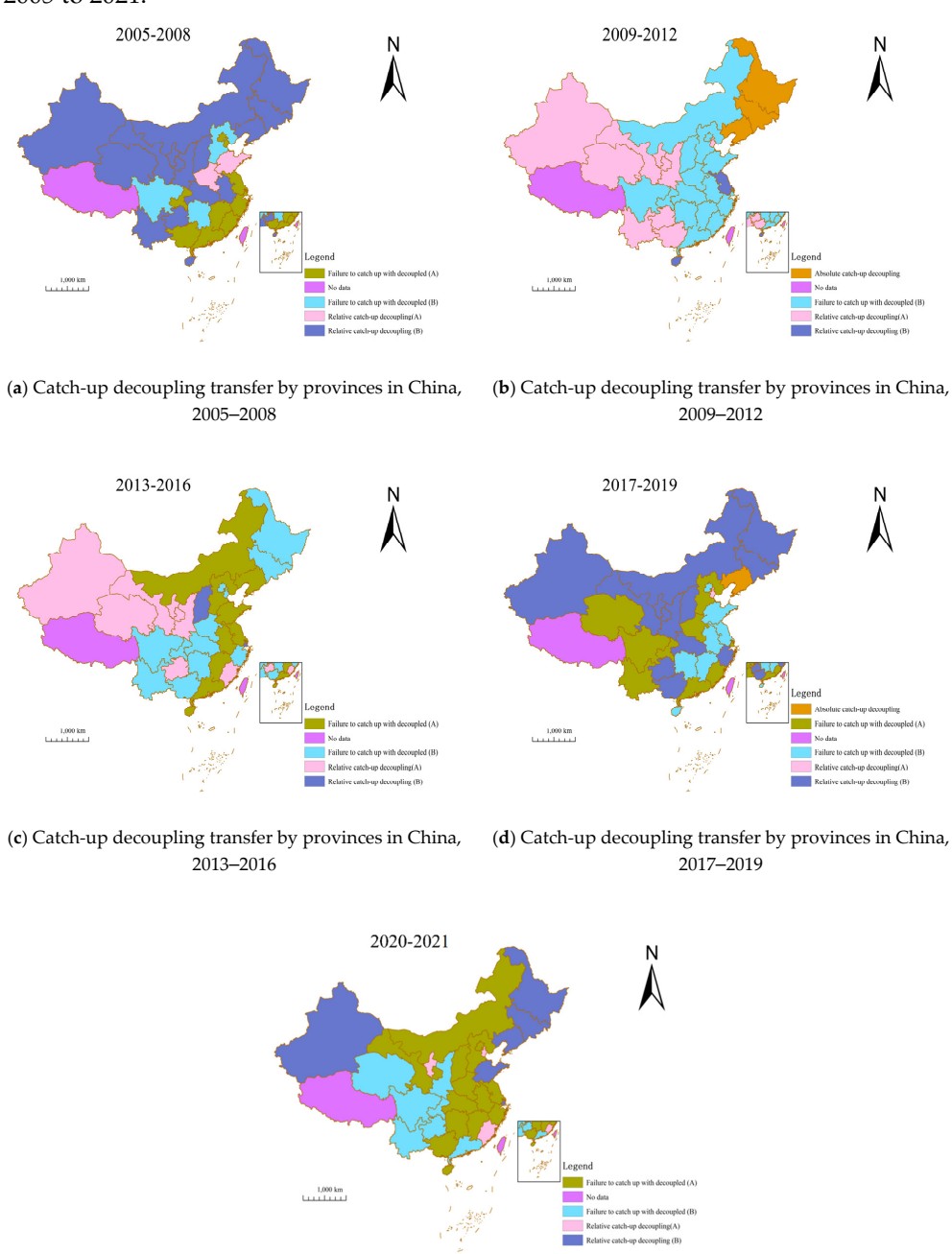

(**a**) Catch-up decoupling transfer by provinces in China, 2005–2008

(**b**) Catch-up decoupling transfer by provinces in China, 2009–2012

(**c**) Catch-up decoupling transfer by provinces in China, 2013–2016

(**d**) Catch-up decoupling transfer by provinces in China, 2017–2019

(**e**) Catch-up decoupling transfer by provinces in China, 2020–2021

**Figure 7.** Evolution of catch-up decoupling status, 2005–2021.

Between 2005 and 2008, most provinces in China were in relative catch-up decoupling B, i.e., the gap between these provinces and the benchmark provinces in terms of energy intensity and per capita GDP levels was gradually narrowing, and catching up in emissions reduction was faster than catching up in the economy. Between 2009 and 2012, the catch-up decoupling status evolved drastically. Three provinces are in absolute catch-up decoupling, 10 provinces are in relative catch-up decoupling A, two provinces are in relative catch-up decoupling B, and 15 provinces are in non-catch-up decoupling B. Three provinces, Heilongjiang, Jilin, and Liaoning, are in relative catch-up decoupling B with the benchmark provinces. $\Delta PG < 0$, which $\Delta CE > 0$, indicating that the economic gap between the three provinces and the benchmark province is getting wider and the carbon emission level is approaching the benchmark province. This result indicates that the three provinces of Heilongjiang, Jilin, and Liaoning were in recession. In 2017–2019, 12 provinces were in relative catching up decoupling B. This indicates that the economic gap and carbon emission gap between the catching up provinces and the benchmark provinces was decreasing, which was due to the fact that the catching up provinces were actively eliminating outdated production capacity, actively introducing advanced production technology, and developing new and high tech industries. In 2020–2021, there was an increase in the number of provinces not catching up and decoupling, i.e., the economic gap between catching-up provinces and benchmark provinces continues to narrow while the carbon intensity gap continues to widen. This result indicates that catching-up provinces are developing their economies at the expense of the environment and energy consumption.

## 4. Conclusions and Policy Recommendations

*Conclusions*

This paper takes the carbon emission from energy consumption as the research object, measures it with cutting-edge measurement methods, analyzes the heterogeneity of China's energy carbon emission in time and space, and introduces the kernel density function, which makes a more intuitive and graphic presentation of the dynamic changes of carbon emission. Based on the log-mean Diels model, the driving factors of energy carbon emissions are comprehensively analyzed. Meanwhile, the decoupling relationship between China's energy carbon emissions and economic development, and the catching-up decoupling are discussed comprehensively, and the main research conclusions are as follows.

Firstly, China's total energy carbon emissions show a "rising-falling-rising" trend, with 2013 as the inflection point and a slight decline in 2021 due to the epidemic.

From the perspective of time dimension, China's total energy carbon emissions from 2005 to 2021 show a trend of "rising-declining-rising", reaching the first peak in 2013, with a total of 1,484,984,406 tons, and the maximum value of 1,605,656 tons in 2020. In 2020, it reaches the maximum value of 16,056,560,017 metric tons in the study period. Subsequently, energy carbon emissions show a slow downward trend. The level of energy carbon emissions in 2021 is almost the same as that in 2019. This is due to the fact that, at the beginning of 2020, China adopted stringent preventive and control measures in order to prevent the spread of the outbreak of Xinguancang epidemic. The short-term social shutdown had a huge impact on the domestic economy and also dampened the growth of carbon emissions.

Secondly, China's energy emissions vary greatly by province, and energy carbon emissions from heavy industrial bases are relatively high.

From the spatial dimension, the energy carbon emissions of China's eastern provinces are larger than those of the central and western regions. Shandong Province's energy carbon emissions ranked first in the country. Shandong's energy structure is dominated by high-carbon fossil energy, with 88% of fossil energy, and fossil energy consumption emits a large amount of carbon dioxide. The central and western regions, such as Qinghai, Ningxia, and Chongqing Municipality, have the lowest energy carbon emissions, and with low levels of economic development, the level of energy consumption is relatively low. Most

other provinces have average annual energy carbon emissions within the range of [50,000, 150,000] million metric tons. Hebei Province, Shanxi Province, and Shandong Province have higher energy carbon emissions. The reason for this is that Shanxi Province is China's energy base, it is particularly rich in coal resources, and its economic growth relies mainly on the large consumption of fossil energy, so energy carbon emissions are relatively high. Hebei Province and Shandong Province are important heavy industry bases in China, and the development of high heavy industry requires the consumption of a large amount of energy, which also leads to energy carbon emissions at the forefront of the country.

Thirdly, China's inter-provincial differences in energy carbon emissions are gradually widening, with an extended trend in the right tail.

In terms of the position of the kernel density function, from 2005 to 2013, the density distribution interval shows an overall rightward shifting trend, indicating that China's energy carbon emissions were continuously rising from 2005 to 2013, which is consistent with the evolution of China's energy carbon emissions in the time dimension. From the kernel function's kurtosis point of view, 2010 shows a double-peak pattern, China's inter-provincial energy carbon emissions are concentrated in 10,000 metric tons as well as 10,000 metric tons, and polarization is more obvious.

Fourthly, the classification of carbon emissions shows an obvious polarization trend, with more than 65% of the provinces belonging to the H-H and L-L categories.

H-H type provinces with high energy carbon emissions and carbon intensity, need to comprehensively consider energy transformation and technological innovation, etc., accelerate energy transformation, and innovate the mode of economic development. H-L type provinces with high energy carbon emissions and low carbon intensity, have economic development at the leading level in China and belong to the economically strong provinces, but their efficiency of energy-use is not high, so these provinces should focus on developing and utilizing clean energy in the future to reduce their dependence on fossil energy. Provinces in the L-H type, such as Jilin, Guizhou, Ningxia, Qinghai, etc., should improve the level of technological innovation, develop the economy, and improve the efficiency of resource utilization.

Fifthly, Contribution of each factor to energy carbon emissions: economic development effect > energy intensity effect > energy structure effect > population size effect.

China's energy carbon emissions are influenced by multiple factors, and the contribution rates of economic development effect, energy intensity effect, energy structure effect, and population scale effect are 863.92%, −446.43%, −81.96% and 58.23% respectively. The overall contribution of the economic development effect to China's energy carbon emissions from 2005 to 2021 is positive, and it is the most important driving force causing for China's energy carbon emissions to continue to climb. The contribution of the population size effect is relatively stable during the study period, and the energy structure has a weak controlling effect on carbon emissions. The inhibitory effect of energy intensity on carbon emissions is gradually weakening, indicating that China still needs to improve the efficiency of energy use and realize the green and low-carbon development of the economy. The contribution of energy structure to carbon emissions is both positive and negative, but overall, the factor is negatively inhibiting the growth of carbon emissions.

Finally, the overall trend of decreasing decoupling richness is shown, and most of the provinces' decoupling status shifts to strong and weak decoupling to each other.

From the perspective of overall decoupling types, there are 4, 3, 3, 4, and 2 decoupling types in the five time periods, respectively, showing a trend of decreasing decoupling richness; from the perspective of inter-provincial decoupling transfer, most of the provinces' decoupling status transfer is dominated by strong and weak decoupling transfer to each other, which indicates that most of China's provinces show a relatively stable decoupling trend of energy and carbon emissions and economic development. From 2020–2022, the provinces that are not been catching up and decoupled will account for a larger share of the total, which means that the economic gap between catching-up provinces and the benchmark provinces with "excellent" economic and carbon emission levels will continue

to narrow, while the gap in carbon intensity will continue to widen. This indicates that, in recent years, China's catching-up provinces have developed their economies at the expense of the environment and energy consumption. The above provinces with decoupled energy and carbon emissions and economic development can be categorized into 2 groups. The first category is economically developed, with a relatively low value-added share of energy industries, such as Beijing and Tianjin. The second category is economically underdeveloped provinces such as Henan and Sichuan, which have achieved the dual purpose of economic growth and environmental protection by taking the road of new industrialization. Provinces showing growth connection, such as Shandong, Hunan, and Guizhou, remain in the crude economic development mode, resulting in synchronized growth of energy and carbon emissions and economy. Provinces such as Ningxia and Xinjiang show negative decoupling of expansion, which is attributed to the fact that in recent years, provinces in western China have vigorously developed coal and other energy-intensive industries in order to develop their economies at the expense of the environment, which has led to a faster rate of energy and carbon emissions than the rate of economic growth.

In view of the above analysis, this paper puts forward the following policy recommendations to reduce China's energy carbon emissions and reach the goal of "carbon peaking and carbon neutrality" at an early date:

(1)　National level

From the previous analysis of the driving factors of China's energy and carbon emissions, the key role is played by the effect of economic development. The main measure of economic development effect is GDP, so to achieve China's 2030 carbon peak and 2060 carbon neutral goals with high quality, taking the sustainable development path is undoubtedly the best choice. Under the current industrial system, technological advances in certain key industrial sectors may reduce carbon emissions and lead to strong green GDP growth.

As China is "rich in coal, low in gas, and short of oil", it has formed an energy structure dominated by coal, and the calorific value of coal is much higher than that of other energy sources, which leads to a large amount of energy and carbon emissions in China, and there is a large gap between the energy and carbon emission efficiencies of developed countries [32]. In view of this, China should deepen the reform of its energy structure, build a stable energy supply system dominated by new energy sources and supplemented by traditional fossil energy sources, and appropriately subsidize clean energy sources at the production and consumption ends. It should also deepen the reform of electric power enterprises, break through the grid bottleneck, and strive to fundamentally solve the bottleneck problem of new energy.

In addition, what affects China's energy carbon emissions is energy intensity. The rate of decline in energy intensity will determine whether China can reach the peak of carbon dioxide emissions by 2030. Therefore, further releasing the energy saving potential and improving the energy utilization rate are the key factors to achieve this goal.

With regard to the population size effect, China, as a large population country, should control the impact of the population size effect on energy and carbon emissions, as China has already liberalized its "three-child" policy. Improve the quality of population, promote the flow of population to energy saving and emission reduction technology research, and accurately match the talent chain around the industrial chain [33]. Rationally plan the development of new urbanization, eliminate the high carbon emissions caused by traditional urbanization, and develop green and locally adapted industries.

China should introduce key technologies and improve efficiency as soon as possible, decarbonize key emission sources, and transition to a green service economy and a low-carbon lifestyle. In addition, China should establish an appropriate low-carbon sustainable development path as soon as possible. This will help mitigate the carbon lock-in effect.

(2)    Provincial level

Most studies on carbon emissions may focus only on the national scale, ignoring the specific contribution of each province. However, some studies have found [34] that it is more effective to study climate change in developing countries using a bottom-up approach. In this sense, this study is able to reflect the "common but differentiated" responsibility for carbon peaking at the provincial level. Based on the calculation and spatial-temporal evolution of carbon emissions in each province of China, a decoupling model is used to explore the relationship between economic development and carbon emissions. The following recommendations are made at the provincial level:

For provinces such as Hebei, Shanxi and Shandong, whose carbon emissions are among the highest in the country, comprehensive consideration needs to be given to aspects such as energy transformation and technological innovation to accelerate energy transformation and innovate economic development. Provinces that are highly likely to peak before 2030, such as Jiangsu, Chongqing, and Shanghai, should prioritize controlling their population growth. For provinces such as Zhejiang and Guangdong, economic development is at a leading level in China and they are strong economic provinces. But the efficiency of energy use is not high. So, these provinces should focus on developing and utilizing clean energy in the future to reduce their dependence on fossil energy. And they should make more efforts to incentivize low-carbon lifestyles and consumption patterns and accelerate renewable energy planning and deployment.

Benchmarking provinces to promote synergistic regional development. There are differences in energy and carbon emissions and levels of economic development among China's provinces. The reason for this is related to the development policies of the Chinese economy. As a result of China's strategy of prioritizing the eastern part of the country, provinces in the central and western regions are lagging behind in terms of economic development. At the same time, western provinces such as Gansu, Xinjiang, and Ningxia still have a large amount of backward production capacity, and the demand for energy is still growing due to the needs of economic development. Provinces should recognize the gap with the benchmark provinces based on the decoupling status of local economic development and carbon emissions and the type of catching up decoupling, based on local resource endowment and development advantages. Try cross-provincial clean energy cooperation, similar to projects like Green Power.

## 5. Limitations and Prospects

As mentioned above, any study of carbon emissions inevitably contains limitations. (1) The calculation of carbon dioxide emissions is somewhat biased. Due to the limitation of data, the carbon dioxide emissions used in this paper are only limited to the carbon dioxide produced by eight common types of energy and electricity consumption. However, we know that in the process of socio-economic development, not only energy consumption produces carbon dioxide, for example, human respiration also produces carbon dioxide. (2) The factors affecting energy carbon emissions are limited. The factors affecting carbon dioxide emissions are large and complex, and this paper only selects four major factors, which lacks comprehensiveness. (3) The study of China's energy carbon emissions in this paper still remains at the national and provincial scales and has not been further disaggregated to the city level due to the difficulty of collecting complete data at the prefecture level. Therefore, in terms of policy recommendations, it is impossible to make carbon emission reduction planning for specific cities. (4) Carbon peak reduction and carbon neutrality are common goals and tasks around the world. In this paper, comparisons with other countries have not yet been discussed.

In the future, we will make efforts in the following areas. (1) Refine the source of carbon dioxide and improve the calculation precision. (2) To make more comprehensive considerations on the factors affecting energy carbon emissions. (3) Utilize the latest international methods to calculate carbon emissions, such as using nighttime lighting data to perform calculations in order to obtain energy carbon emissions at the prefecture and

municipal levels. (4) Introduce the carbon emissions of other countries in the international arena for comparison with China. Scientifically formulate policies to mitigate global warming. Promote the sustainable development of human beings.

**Author Contributions:** Conceptualization, J.J. and J.L.; methodology, J.J.; software, C.L.; validation, J.J., Y.S. and X.Y.; formal analysis, J.J.; resources, X.Y.; data curation, C.L.; writing—original draft preparation, J.J.; writing—review and editing, J.J.; visualization, Y.S.; supervision, J.L.; project administration, J.L.; funding acquisition, J.L. All authors have read and agreed to the published version of the manuscript.

**Funding:** This research was funded by General Program of the National Social Science Foundation, grant number 21BGL166 and Innovation and Entrepreneurship Training for Students of Northeast Forestry University, grant number 202310225250.

**Institutional Review Board Statement:** Not applicable.

**Informed Consent Statement:** Not applicable.

**Data Availability Statement:** Data are mainly from the China Statistical Yearbook and the China Energy Statistical Yearbook.

**Conflicts of Interest:** The authors declare no conflict of interest.

## Appendix A

**Table A1.** The energy carbon emission calculations for all provinces in China from 2005 to 2021.

| Region / Year | 2005 | 2006 | 2007 | 2008 | 2009 | 2010 | 2011 | 2012 | 2013 | 2014 | 2015 | 2016 | 2017 | 2018 | 2019 | 2020 | 2021 |
|---|---|---|---|---|---|---|---|---|---|---|---|---|---|---|---|---|---|
| Beijing | 11,998.62 | 12,327.388 | 13,113.708 | 13,336.661 | 13,679.251 | 13,867.019 | 12,932.269 | 13,139.232 | 12,074.419 | 12,511.52 | 12,144.764 | 11,496.04 | 11,288.46 | 11,565.706 | 11,524.71 | 11,654.88 | 12,568.55 |
| Tianjin | 12,488.522 | 13,293.075 | 14,127.898 | 14,055.932 | 15,107.56 | 18,436.843 | 20,231.116 | 20,377.362 | 21,142.105 | 20,419.87 | 20,158.968 | 19,033.916 | 18,905.176 | 19,623.763 | 19,788.62 | 20,147.33 | 21,456.44 |
| Hebei | 57,412.516 | 61,942.581 | 67,607.595 | 70,642.242 | 75,350.813 | 81,068.028 | 91,675.929 | 92,966.828 | 93,243.415 | 88,685.007 | 92,733.758 | 92,865.965 | 92,193.492 | 94,559.178 | 94,913.14 | 10,4878.33 | 106,589.12 |
| Shanxi | 56,159.395 | 62,219.48 | 64,241.45 | 63,124.779 | 62,530.73 | 67,088.454 | 73,955.093 | 77,225.595 | 79,194.815 | 81,095.64 | 93,090.316 | 92,133.059 | 97,087.731 | 103,926.66 | 109,327.2 | 115,200.55 | 125,633.31 |
| Inner Mongolia | 31,179.32 | 36,618.267 | 42,270.083 | 50,463.686 | 54,940.466 | 60,551.772 | 75,579.624 | 78,531.547 | 76,723.589 | 78,612.072 | 77,969.182 | 78,887.758 | 83,004.003 | 95,196.635 | 105,600.2 | 115,268.14 | 123,456.22 |
| Liaoning | 49,813.984 | 53,546.913 | 57,952.996 | 59,414.946 | 61,512.03 | 67,401.813 | 71,990.851 | 74,604.894 | 71,921.114 | 72,027.084 | 69,545.834 | 70,442.413 | 72,570.98 | 77,177.94 | 84,066.75 | 94,066.755 | 96,541.36 |
| Jilin | 18,427.643 | 20,086.88 | 21,219.444 | 22,019.111 | 22,525.58 | 25,131.01 | 28,754.483 | 28,415.645 | 27,380.831 | 27,158.152 | 23,184.056 | 22,859.128 | 22,682.756 | 23,468.747 | 24,176.49 | 25,796.34 | 26,896.85 |
| Heilongjiang | 25,149.453 | 26,565.226 | 28,570.084 | 30,305.328 | 31,646.141 | 34,319.075 | 36,805.972 | 38,555.263 | 36,586.454 | 37,082.37 | 33,939.679 | 34,224.632 | 34,248.898 | 34,931.633 | 36,671.46 | 37,898.666 | 38,744.223 |
| Shanghai | 23,128.306 | 23,039.481 | 23,657.101 | 24,791.424 | 24,681.47 | 27,004.166 | 27,780.521 | 27,389.171 | 29,052.531 | 26,454.763 | 26,564.044 | 26,471.325 | 26,967.67 | 26,550.202 | 27,403.58 | 29,635.224 | 29,888.22 |
| Jiangsu | 47,340.372 | 51,815.725 | 55,854.135 | 57,632.443 | 60,246.025 | 67,259.233 | 77,527.449 | 79,158.29 | 81,395.446 | 80,822.449 | 83,706.697 | 87,125.359 | 86,307.475 | 85,537.563 | 87,372.58 | 88,746.32 | 92,746.32 |
| Zhejiang | 29,982.72 | 33,830.376 | 37,798.321 | 38,546.641 | 40,080.3 | 43,024.865 | 45,569.253 | 44,210.237 | 45,473.688 | 44,952.697 | 45,583.628 | 45,243.106 | 47,384.075 | 46,475.895 | 47,476.50 | 48,746.369 | 49,746.369 |
| Anhui | 19,693.454 | 21,189.638 | 23,602.034 | 26,945.052 | 29,653.55 | 31,428.394 | 34,147.836 | 35,238.853 | 38,271.91 | 39,541.489 | 39,619.898 | 39,553.309 | 40,938.104 | 42,504.565 | 42,779.28 | 43,779.285 | 44,779.285 |
| Fujian | 13,451.65 | 14,719.621 | 16,501.447 | 17,262.755 | 20,448.583 | 22,721.407 | 25,938.656 | 25,739.451 | 25,273.295 | 28,834.236 | 27,727.986 | 26,018.475 | 27,412.794 | 30,337.484 | 32,290.43 | 34,521.356 | 38,521.356 |
| Jiangxi | 11,730.639 | 12,799.931 | 13,982.176 | 14,209.29 | 14,875.145 | 17,300.321 | 23,669.643 | 19,208.941 | 20,697.428 | 21,081.452 | 21,969.719 | 22,259.928 | 22,729.037 | 23,758.601 | 24,266.73 | 26,456.369 | 29,456.369 |
| Shandong | 70,074.78 | 78,711.097 | 87,506.502 | 94,495.691 | 98,401.375 | 108,777.02 | 110,085.24 | 120,622.48 | 117,437.23 | 125,785.31 | 138,498.17 | 145,505.03 | 149,306.95 | 147,839.73 | 151,698.2 | 174,596.33 | 194,596.33 |
| Henan | 42,521.633 | 48,219.488 | 53,375.018 | 55,077.511 | 56,281.441 | 60,866.511 | 67,090.084 | 62,784.57 | 62,311.133 | 62,987.968 | 59,313.762 | 58,626.372 | 57,379.914 | 57,717.863 | 53,398.49 | 60,245.734 | 62,245.734 |
| Hubei | 23,951.237 | 26,514.086 | 29,337.711 | 29,358.838 | 31,527.241 | 36,156.131 | 41,145.958 | 41,171.968 | 35,834.787 | 36,282.25 | 34,444.153 | 34,258.002 | 35,062.216 | 36,458.122 | 38,710.66 | 39,443.35 | 42,443.35 |
| Hunan | 22,660.181 | 24,173.136 | 26,557.418 | 26,295.89 | 27,656.68 | 29,342.556 | 32,755.442 | 32,264.79 | 31,382.114 | 30,470.505 | 30,348.687 | 31,037.192 | 31,337.817 | 32,037.033 | 31,947.22 | 33,467.521 | 36,467.521 |
| Guangdong | 38,711.841 | 43,109.337 | 46,767.907 | 48,240.277 | 52,047.24 | 57,628.255 | 63,557.26 | 61,867.836 | 62,952.558 | 63,402.06 | 63,996.629 | 66,114.53 | 69,203.562 | 71,499.158 | 70,979.04 | 78,465 | 82,465 |
| Guangxi | 10,261.545 | 11,265.167 | 12,868.063 | 12,757.037 | 14,148.72 | 17,212.221 | 21,173.247 | 23,254.117 | 23,380.218 | 23,210.166 | 21,791.625 | 22,655.55 | 23,965.271 | 25,181.685 | 26,628.65 | 27,728.365 | 28,728.223 |
| Hainan | 1598.842 | 2428.628 | 4409.141 | 4671.755 | 4966.895 | 5423.367 | 6392.068 | 6662.906 | 6190.831 | 6849.103 | 7498.034 | 7278.204 | 7070.089 | 7482.044 | 7685.407 | 7885.239 | 8085.698 |
| Chongqing | 9068.654 | 9834.204 | 10,715.227 | 13,268.907 | 14,269.873 | 15,754.548 | 17,945.746 | 17,713.05 | 15,341.73 | 16,441.897 | 14,420.451 | 14,862.151 | 15,321.748 | 15,422.782 | 15,565.26 | 16,565.665 | 18,565.333 |
| Sichuan | 21,609.477 | 24,174.667 | 27,013.884 | 29,831.582 | 33,567.332 | 34,594.562 | 34,904.466 | 36,346.76 | 37,329.405 | 38,653.411 | 33,105.502 | 32,553.343 | 31,964.3 | 31,171.36 | 33,463.12 | 34,463.887 | 36,463.635 |
| Guizhou | 18,222.234 | 21,208.949 | 22,687.785 | 21,057.055 | 23,065.929 | 23,247.165 | 25,705.446 | 28,128.194 | 29,204.301 | 28,206.238 | 28,228.924 | 29,571.299 | 29,741.36 | 27,410.37 | 28,084.69 | 29,410.987 | 32,410.336 |
| Yunnan | 17,688.943 | 19,464.585 | 20,336.834 | 20,922.611 | 22,713.016 | 23,979.158 | 24,755.196 | 25,707.336 | 25,436.331 | 22,871.815 | 20,711.095 | 20,486.29 | 21,760.629 | 24,186.349 | 25,297.39 | 26,300.446 | 29,300.332 |
| Shaanxi | 17,646.677 | 21,437.586 | 23,661.633 | 26,659.738 | 28,966.349 | 34,307.235 | 37,930.583 | 43,553.726 | 46,265.637 | 48,745.6 | 48,264.732 | 49,160.574 | 50,667.473 | 49,574.909 | 53,839.69 | 57,555.338 | 60,555.225 |
| Gansu | 12,953.157 | 13,819.341 | 15,370.926 | 15,672.057 | 15,476.926 | 17,227.728 | 19,902.132 | 20,497.592 | 21,197.156 | 21,348.103 | 20,648.926 | 19,863.062 | 20,038.48 | 20,992.541 | 21,253.85 | 26,666.125 | 27,666.898 |
| Qinghai | 2344.101 | 2944.23 | 3266.284 | 4067.184 | 4142.595 | 4133.085 | 4885.542 | 5824.752 | 6414.459 | 5985.861 | 5525.828 | 6435.889 | 6167.295 | 6032.492 | 5963.033 | 6422.001 | 7422.456 |
| Ningxia | 7310.221 | 8008.046 | 9057.795 | 10,009.469 | 11,008.736 | 13,027.29 | 17,343.634 | 18,634.516 | 19,829.671 | 20,184.71 | 20,957.493 | 20,860.469 | 25,644.47 | 28,583.99 | 31,066.64 | 35,100.045 | 38,100.693 |
| Xinjiang | 15,402.079 | 17,550.824 | 19,107.481 | 21,153.436 | 24,690.738 | 27,656.497 | 32,715.645 | 37,794.645 | 43,279.765 | 48,110.51 | 49,709.02 | 51,870.247 | 55,171.86 | 57,433.832 | 61,079.01 | 63,000.123 | 65,055.361 |

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
