# Peer review of "Analysis of the Spatial and Temporal Evolution of China’s Energy Carbon Emissions, Driving Mechanisms, and Decoupling Levels"

_sustainability, doi:10.3390/su152215843_

Round 1
Reviewer 1 Report
Comments and Suggestions for Authors
The paper contains so many technical elements that are not familiar to readers of the Sustainability journal, and we suggest submitting the article to another journal that fits with the content of the article.
The abstract section shifts the focus of the research study by emphasizing different variables, which is confusing and does not attract a reader to read further, as the sentence structure needs careful re-structuring and re-writing. The research ideas could have been more effective through elaborative and concise sentences. The abstract does not provide a brief account of the work and conclusion of the research study. It needs to be more structured and synthesized for research clarity. The manuscript lacks a structured research methodology.
The Materials and Methods section looks like a puzzle with much information but without a relationship. In this section, the authors should add more details regarding the research data collection, analysis, and interpretation of results. They should also briefly describe the methods of data employed and their application and appropriateness for data analysis.
The results were not well-presented to readers to understand the focus of the research study.
The results must be interpretive rather than just descriptive and connect the research results with relevant literature citations for validity and reliability.
The research data does not support the conclusions and policy recommendations section, which does not indicate a more straightforward path for future studies.
Figure 3 and Figure 4 are missing. After Figure 2, the following figure is Figure 5.
The sources of the figures are missing.
Not all figures are introduced in the text before they are presented.
For example, Figure 2 Evolutionary characteristics of China's energy carbon emissions, 2005-2021
The seven figures are so many and are not critically analyzed.
The tables are not discussed in a relationship with the paper's objectives.
Comments on the Quality of English LanguageThe structure of the sentences should be improved to logical flow and to avoid segmentation of the information.
Reviewer 2 Report
Comments and Suggestions for Authors
This is an interesting study to use mathematical models to analyze and predict the China's energy carbon emissions, and also provide some analysis about its intrinsic mechanisms toward a sustainable development goal.
Authors may consider below comments
1) Besides the energy carbon emission, better to quantitatively measure the CO2 emissions in terms of Metric tons /Yr accordingly to the international standards, eg. Figure 1.
2) CAN Maka a comparison with other countries such as Brazil, India or even Europe? either citing references or getting basic data from other countries' statistics data
3) The section 4 conclusion and policy recommendations may needs to be re-stated clearly with possible action plan
Comments on the Quality of English Language
Good to have Polishing
Reviewer 3 Report
Comments and Suggestions for Authors
Dear author,
i suggest several improvements.
Review: Analysis of the spatial and temporal evolution of China's energy carbon emissions, driving mechanisms and decoupling levels
Dear authors, the paper focus an important topic as pollution. China is a very important case study on this topic that must be monitored. Nevertheless, the paper must be revised. I suggest the following improvements:
- In the text i suggest writing CO2 as subscript and not CO2 or CO2;
- The refences number does not respect the MDPI template, I suggest writing them as [1], [2] ……
- Often after the dot the space does not used, for example: lines 131 (.EFi), lines 130 source’sNCVi
- An error occurs in the sentence from line 131 to 133, I suggest reviewing it
- The sentence from line 156 to 159 are not clear, I suggest reviewing it.
- Line 161 there is not the dot before Accordingly.
- Paragraph 2.5 presents several mistakes as listed above.
- Lines 257, 261: I suggest writing the tons in correct formula. See MDPI suggestions.
- Paragraph 3.3: I suggest modifying the bullet list from (1) to (4). These numbers could be linked to the above equations.
- Paragraph 4 I suggest modifying the bullet list from (1) to (4). These numbers could be linked to the above equations.
Comments on the Quality of English Languagemoderate editing required
Reviewer 4 Report
Comments and Suggestions for Authors
The largest absolute national contribution to global CO2 emissions now is from China, which currently accounts for 30% of global emissions. Like many other countries, the primary cause of anthropogenic CO2 emissions is energy-related fossil fuel combustion. China’s 1.4 billion people consume energy to meet their daily needs, including heating and cooling of their living and working places, fuel for cooking their meals, electricity to power their appliances and equipment, and fuels for both their own personal transportation as well as the products they purchase. China's strategy is aimed at enhancing energy conservation, optimizing the energy structure, and supporting the development of nuclear energy and renewables. With these efforts, the increase in energy-related CO2 emissions has been reduced. However, future emissions still need to be decreased significantly, and the drivers of emissions as well as the political and technological drivers of the reduced total and relative (per capita or gross domestic product [GDP]) CO2 emissions need to be explored, quantified, and better understood. Within this context, the present study aims to measure the carbon emissions and carbon intensity of energy consumption in China, and to explore regional differences using the carbon emission classification method and the kernel density index method. The results show that China's energy carbon emissions are affected by multiple factors, and the contribution of each factor to energy carbon emissions is ranked as follows: economic development effect>energy intensity effect>energy structure effect>population size effect of provinces over 65%.
Generally, some revision suggestions are listed below:
1. There is a major problem based on the 2005-2021 data. The steep drop in carbon emissions in 2020-2021 is due to the impact of the "new crown" epidemic since 2020. Thus, the sudden change after 2020 is not comparable with the continuity from 2005-2019.
2. The study needs to add a certain process of verification and comparative analysis of the results of emission calculations to ensure the accuracy of the study.
3. Add a conclusion section to discuss the practical significance of the study and the prospect of future research related to the description.
Also, please use scientific notation in your articles for clarity and ease of understanding
Overall, the topic selection of this paper is not innovative, and there are already many related studies, and the research methods are all conventional, without outstanding innovation.
Reviewer 5 Report
Comments and Suggestions for Authors
The study has investigated the temporal and spatial evolution of energy carbon propagation in China.
The work is novel and interesting, specifically the part on the dynamic decoupling process instead of a static approach.
The method, results, and conclusion are well-organized and well-described.
Hence, the paper is suitable for publication.
Author Response
Dear reviewer.
Thank you very much for taking your valuable time to review this manuscript. I appreciate your recognition and compliments. I wish you a wonderful day.
Yours sincerely, Ji Jingyi
Corresponding author, Jiehua Lv
Round 2
Reviewer 1 Report
Comments and Suggestions for Authors
Good luck!
Comments on the Quality of English LanguageMinor revisions to the structure of the sentences are required.
Author Response
Dear reviewer,
Thank you for your sincere advice. Your suggestions were really very helpful to us and made our article much improved. I appreciate it very much and thank you for your sincere and valuable comments. We have tried our best to improve the English sentences and structure of the article. Thank you for your hard work. I wish you all the best.
Yours sincerely,Jingyi Ji
Reviewer 3 Report
Comments and Suggestions for Authors
Dear authors, the paper has been revised but I suggest the following improvements, see the attached file.

Author Response
Dear reviewers,
Thank you for your sincere and professional advice. We have changed the bullet list from (1) to (4) without brackets. And updated it in our latest manuscript. We have carefully checked the references and hope that there will be no more errors in the text. Thank you for your hard work. I wish you all the best.
Yours sincerely,Jingyi Ji

Reviewer 4 Report
Comments and Suggestions for Authors
The article has been substantially revised. Accept in present form.
Author Response
Dear reviewers,
Thank you for your sincere and professional advice. Thank you for your hard work. Wish you all the best.
Yours sincerely,Jingyi Ji